# Ensuring Academic Integrity and Trust in Online Learning Environments: A Longitudinal Study of an AI-Centered Proctoring System in Tertiary Educational Institutions

**Christos A. Fidas [1,*]**, **Marios Belk [2]**, **Argyris Constantinides [3]**, **David Portugal [4]**, **Pedro Martins [4]**, **Anna Maria Pietron [2]**, **Andreas Pitsillides [3,5]** and **Nikolaos Avouris [1]**

[1] Department of Electrical and Computer Engineering, University of Patras, 26504 Rio, Greece
[2] Cognitive UX GmbH, 69253 Heiligkreuzsteinach, Germany; belk@cognitiveux.com (M.B.); anna.pietron@cognitiveux.de (A.M.P.)
[3] Department of Computer Science, University of Cyprus, Nicosia 2109, Cyprus; constantinides.argyris@ucy.ac.cy (A.C.); andreas.pitsillides@ucy.ac.cy (A.P.)
[4] Institute of Systems and Robotics, University of Coimbra, 3030-290 Coimbra, Portugal
[5] Department of Electrical and Electronic Engineering Science, University of Johannesburg, Auckland Park P.O. Box 524, South Africa
[*] Correspondence: fidas@upatras.gr

**Abstract:** The credibility of online examinations in Higher Education is hardened by numerous factors and use-case scenarios. This paper reports on a longitudinal study, that spanned over eighteen months, in which various stakeholders from three European Higher Education Institutions (HEIs) participated, aiming to identify core threat scenarios experienced during online examinations, and to, accordingly, propose threat models, data metrics and countermeasure features that HEI learning management systems can embrace to address the identified threat scenarios. We also report on a feasibility study of an open-source intelligent and continuous student identity management system, namely TRUSTID, which implements the identified data metrics and countermeasures. A user evaluation with HEI students ($n = 133$) revealed that the TRUSTID system is resilient and effective against impersonation attacks, based on intelligent face and voice identification mechanisms, and scored well in usability and user experience. Aspects concerning the preservation of privacy in storing, retrieving and processing sensitive personal data are also discussed.

**Keywords:** online academic activities; learning management systems; perceived credibility; user survey; specifications analysis

## 1. Introduction

Recent advancements in information and communication technologies necessitated a rapid transition in Higher Education Institutions (HEIs) towards completely online academic teaching, learning and examination paradigms [1–8]. In most cases, this was accomplished by utilizing Video Conferencing Tools (VCTs), in combination with Learning Management Systems (LMSs), with the aim to continue academic activities, like teaching, laboratory work and examinations, within an online context.

Nonetheless, the aforementioned transition faced several challenges in ensuring the satisfactory implementation of HEI curricula and fair student evaluation processes so as to sustain credibility in online academic activities. In this context, a key challenge related to the establishment of appropriate procedures for identifying non-permitted activities from students, such as prohibited communication and collaboration among students, and, also, impersonation cases (i.e., intentionally pretending to have someone's identity in order to unethically participate in academic activities).

To this end, many online-transitioned HEIs encountered difficulties in effectively identifying such malicious activities. The adopted LMS and/or VCT tools entail a single

entry-point for student authentication and identification as a precondition for participation in online academic activities and to gain access to protected educational material and services. With regards to the LMS, students typically authenticated through a single-point textual password system, which assumed integrity of the student's attendance within the whole academic activity. With regards to VCT, student identification and verification procedures were primarily conducted manually through human intervention (e.g., instructors, invigilators). The aforementioned approaches fall short in detecting fraudulent student activities after the single entry point of authentication is successfully performed [9]. In addition, manual student identification is a time consuming and difficult endeavor for instructors throughout the whole academic activity.

To alleviate these issues, numerous works on continuous or implicit authentication methods have been proposed as an additional non-intrusive security countermeasure [9–15]. However, existing solutions that simply monitor face and/or body cues are not adequate to prevent fraudulent behavior in online examinations, since they lack student interactions and are not able to capture scenarios in which the camera stream switches over to other video sources [16]. Other works focused on the combination of multiple biometric traits [17,18], such as face with fingerprint [19], fingerprint with vocal traits [20], and fingerprint with mouse patterns [21]. Nevertheless, multiple biometric solutions often operate in an intrusive fashion that interferes with student activities and they require additional devices [21]. The use of behavioral biometrics was investigated by analyzing keystroke and mouse patterns, while students took online examinations, in order to provide transparent authentication [22,23]. However, such behaviorally based approaches are often unreliable without the combination of a physical biometric trait and are usually limited to specific interaction types (e.g., recognizing keystroke patterns is effective only when the student is typing) [16].

Considering the importance of tackling the threat of impostors that intentionally adopt another student's identity in the maintenance of credibility in online activities, it is evident that online-transitioned education systems must be able to continuously verify student presence and attendance in several educational activities of critical importance (e.g., laboratory quizzes and online examinations). Nevertheless, existing solutions are usually designed only as proctoring tools during online examinations with a person monitoring students remotely (e.g., Kryterion (https://www.kryterion.com, accessed on 26 May 2023), ProctorU (https://www.proctoru.com, accessed on 26 May 2023)). Such tools tend not to consider participation in the rest of the course, nor other important threats related to communication and collaboration with other persons and access to resources. Furthermore, they are not scalable [16], require technological infrastructure and setup [24], and fall short in addressing privacy concerns with regards to recorded videos [24–26]. Other cost-effective proctoring solutions aim to either eliminate proctor efforts, by replacing people with algorithms that conduct post-analysis or real-time analysis (e.g., ProctorFree (https://www.proctorfree.com, accessed on 26 May 2023), SMOWL (https://smowl.net/en, accessed on 26 May 2023), Respondus (https://web.respondus.com, accessed on 26 May 2023)), or utilize hybrid solutions that involve people only in cases of suspicious behaviors (e.g., Proctorio (https://proctorio.com, accessed on 26 May 2023), [27]). However, such solutions might entail numerous risks, due to the variability of algorithm accuracy and the limited scenarios supported [16]. In addition, recent research on existing proctoring solutions tend to be sparse and question the effectiveness of proctoring in terms of reducing cheating scenarios and academic misconduct [28].

**Research Motivation and Contributions.** Based on the aforementioned works, several efforts have addressed existing challenges and issues related to student identification, verification and interaction behavior within critical online academic activities. However, these solutions mostly favor a certain user feature within a certain interaction system, when, in fact, solutions combining multiple sources of input (e.g., face, voice, interaction behavior) under an agile system integration framework, are required. So far, to the best of the authors' knowledge, no other solution incorporates a system integration framework, bootstrapped

on the synchronous and asynchronous critical online academic activities of HEIs, as well as on different examination types and procedures.

The main contributions of the paper are as follows: *(i)* we provide important insights from HEI stakeholders (i.e., HEI instructors, students, administrators) on their timely experiences of, and perceived credibility of, online examinations and other critical academic activities over the COVID-19 period; *(ii)* we identify and compile a comprehensive list of threat scenarios, based on feedback received from relevant HEI stakeholders; *(iii)* we suggest countermeasures to overcome the identified threat scenarios, based on feedback received from field experts (i.e., software engineers, administrators, HEI instructors); *(iv)* we implement a series of countermeasures within an open-source software prototype, bootstrapped on synchronous and asynchronous critical online academic activities and on different examination types and procedures of HEIs; and *(v)* we provide insights from a preliminary user evaluation of the software prototype, which is an important step towards realizing a comprehensive and holistic proctoring system within the Higher Education domain.

This paper is structured as follows. In the next section we present the method of study, research questions, procedure and analysis of user responses of two subsequent user studies, including key threat scenarios that were identified based on the analysis. Next, we present the countermeasures and features by which future learning management systems may address the identified threat scenarios. Consequently, we present the implementation details of an intelligent and continuous online student identity management framework, namely TRUSTID, followed by user evaluation of the implemented framework. Finally, we present state-of-the-art architectural solutions for preservation of data privacy, and conclude the paper with the limitations of this work.

## 2. Method and Context of the Study

### 2.1. Research Methodology

The research methodology was split into three main phases. Specifically, as part of *Phase A*, we conducted a needs analysis on how stakeholders from Higher Education Institutions (HEIs) perceived the credibility of critical academic activities as experienced during the COVID-19 period and identified threat scenarios and malicious student activities during critical academic activities. Aiming to increase ecological validity, we conducted this research study slightly after the COVID-19 period to gain insights from participants that had timely real-life experiences with regards to the research questions that were under investigation. A total of thirty-one (31) individuals having varying backgrounds, namely, students, instructors, system administrators, decision makers, and data protection experts, from three European universities *(University of Patras, Greece; University of Cyprus, Cyprus; University of Coimbra, Portugal)* participated. Each participant took part in semi-structured interviews that lasted approximately 60–90 min each, discussing different topics based on the research questions set as part of this phase.

In *Phase B* we conducted a needs verification analysis to verify, with HEI stakeholders, the outcome of the needs analysis, we rated the identified threat scenarios in terms of the likelihood/probability of occurrence and level of severity, and we identified challenges for adoption and relevant privacy issues. A total of seven (7) individuals participated from the same three European universities, having the same profiles as those in Phase A. In a similar manner to that of Phase A, we conducted a series of semi-structured interviews with each participant that lasted approximately 60–90 min each, in which each participant verified the outcome of the needs analysis and rated the identified threat scenarios in terms of likelihood/probability of occurrence and level of severity.

Finally, in *Phase C* we identified countermeasures and proposed features that a framework could implement to address the identified threat scenarios during critical academic activities. In particular, by means of behavioral and contextual data analyses, we identified countermeasures related to student verification during enrolment, and continuous student identification during an examination, based on biometric data analysis (physiological and behavioral) and monitoring of student behaviors and physical and digital contexts. We

also propose state-of-the-art architectural solutions to address aspects of the preservation of privacy in the storing, retrieving and processing of sensitive personal student data.

### 2.2. Background of Participating HEIs

**Adopted Information and Communication Technologies (ICT) Solutions.** All participating HEIs adopted ad-hoc ICT solutions during COVID-19, based on the following two-step method: *(a)* HEIs utilized existing off-the-shelf Video Conferencing Tools (VCTs) aiming to communicate in real-time with their students; and *(b)* HEIs utilized existing Learning Management Systems (LMSs) for content sharing and asynchronous communication. The deployed VCTs and LMSs were deployed as independent tools, which, however, were used simultaneously for critical academic activities within online contexts.

With regards to VCTs, there is a common approach in using off-the-shelf tools like Zoom, Microsoft Teams, Skype for Business, etc., as part of both teaching and examination activities. All HEIs continued using the LMSs they had been using prior to the COVID-19 period, mainly due to familiarity aspects. In particular, one university utilized a nation-wide LMS, another university utilized an in-house developed LMS, while the third university utilized off-the-shelf/open-source LMSs, such as Moodle and Blackboard.

**Examination Types and Modalities.** The participating HEIs utilized the following three (3) different online examination types, which entail different contextual modalities and characteristics: oral online examinations, digital written online examinations, and hand written online examinations (see Table 1 for details).

**Table 1.** Examination modalities and contextual characteristics per examination type.

| Online Examination Type | Modalities | Contextual Characteristics |
|---|---|---|
| Oral Online Examination | Instructor asks real-time questions or shares questions (picture, diagram, etc.) through a conference system. Then, each student provides the answer orally. | Time constraints for providing each answer. Examination classrooms usually have a limited number of students to be examined (e.g., up to five). |
| Digital Written Online Examination | Instructor shares the examination questions through the LMS. Students login to the LMS and either view the questions (e.g., multiple-choice questions) and directly provide answers to each question through the LMS, or download a document with questions and further upload their answers to the LMS. Students typically utilize a computer keyboard and computer mouse creating keystroke and computer mouse movement input streams. | Time constraints typically apply for the whole examination session. In some cases, time constraints may be applied for the provision of each answer. Examination classrooms do not have limitations with regards to the number of students attending. Instructors use conferencing systems to monitor between 30 to 70 students simultaneously within a virtual classroom. Direct audiovisual communication with a certain student is performed, when necessary, through the conferencing system. |
| Hand Written Online Examination | Instructor shares the examination questions through the LMS (usually as a PDF). Then, each student either views the questions through the LMS or downloads the PDF on his or her computer. Student writes the answers on hard copy sheets and, finally, uploads the hard copy sheets to the LMS. | Time constraints typically apply for the whole examination, or for each question. Instructors use conferencing systems to monitor between 30 to 70 students simultaneously within a virtual classroom. Direct audiovisual communication with a certain student is performed, when necessary, through the conferencing system. |

## 3. Insights on Stakeholders' Views and Experiences in Applying Online Examinations

This phase of the user study aimed to gain insights on HEI stakeholders' experiences with regards to online examinations and other critical academic activities during

the COVID-19 period, in order to elicit their perceived credibility and to identify threat scenarios affecting the credibility of such activities.

### 3.1. Research Questions

We investigated using the following research questions:

*(RQ1)* How did the stakeholders perceive the credibility of online examinations and other critical academic activities during the COVID-19 period (March 2020–August 2021)?

*(RQ2)* Which threat scenarios that include impersonation activities of students should be urgently addressed during online examinations?

*(RQ3)* Which threat scenarios that include forbidden communication, collaboration and/or resource access activities of students should be urgently addressed during online examinations?

### 3.2. Sampling and Procedure

A total of 31 participants were recruited from three European universities. Table 2 summarizes the number of participants per stakeholder group and participating Higher Education Institution. The sample included participants with a variety of roles, i.e., students, instructors, system administrators, decision makers and data protection experts. We conducted a series of semi-structured interviews with each participant that lasted approximately 60 to 90 min each, discussing different topics based on the research questions.

**Table 2.** Number of participants per stakeholder group and participating Higher Education Institution.

| Stakeholder Group | Higher Education Institution 1 | Higher Education Institution 2 | Higher Education Institution 3 |
|---|---|---|---|
| Students | 2 | 3 | 3 |
| Instructors | 3 | 4 | 3 |
| System Administrators | 2 | 2 | 2 |
| Decision Makers | 2 | 1 | 1 |
| Data Protection Experts | 1 | 1 | 1 |
| **Total** | **10** | **11** | **10** |

### 3.3. Analysis of Results

3.3.1. Perceived Credibility of Online Examinations (*RQ1*)

In order to examine answers to *RQ1*, we asked participants a series of questions to elicit their perceived credibility of online academic activities within the current LMSs at their universities. Appendix A provides and describes the guide for the stakeholder interviews, including the interview discussion themes and questions. The questions of the interview guide were designed by consulting relevant experts in the field (academic instructors, system administrators in HEIs, data protection experts) from three European Universities.

We asked participants to report their best and worst experiences with regards to critical online academic activities (e.g., examinations, laboratory work). The majority of instructors expressed their concerns about the limitations of current practices and technologies within HEIs. In addition, they underpinned the necessity to seek new technologies to overcome the challenges of trustworthiness in online examinations.

Representative responses from participants included the following: *"I have lost my trust to the ability of my university to fulfill its mission, which is to assure that students have acquired the necessary knowledge and skills needed for their profession. The current procedures and workflows in which the online examinations, during COVID-19, take place are questionable in terms of credibility"*—Instructor 2; *"No contact with the attendees, no clue of what's happening with the students, who is with them, whether they communicate with others through the web or other tools"*—Instructor 1; *"Stressful because of inexistent trustworthiness and lack of 'weapons' to fight the problem"*—Instructor 2; *"During the online examinations, I was really stressed because I did not have control about what was going on in each student's physical space, whether they received*

*any help from others"*—Instructor 3; *"Ambiguities about the trustworthiness of conducting online exams"*—Instructor 4; *"In the online learning activities, students can join and leave without the instructor noticing them. There are also cases in which students join the call and they 'appear to be connected' but are not really there. This was evident from cases in which I asked students to type in their name in the chat and many of the 'connected users' did not type it"*—Instructor 5; *"There were a lot of problems with the current technologies at the university. People were complaining especially in the beginning and we had several issues with the video server, we had to handle all the network issues, missing events, real-time communication etc. It was really difficult to make 'everyone happy' and compromises were made to make sure everybody was served"*—System Administrator 1; *"Our systems were not ready for online examinations. The bad experience was preparing LMS to handle the new increased volume of usage. I had to set up a new upgraded server to support examinations through a particular LMS"*—System Administrator 2.

We asked participants to further elaborate whether they believed that the current procedures at their organizations were trustworthy with regards to critical online academic activities.

Representative responses from participants included the following: *"I had several exams online and there was really no way of controlling what we were doing in those, so it's not trustworthy at all, because I was talking to some of my colleagues, and we were discussing the exam, although the teachers said that we cannot talk to our colleagues"*—Student 4; *"There are companies that provide personalized exam solutions (e.g., pass the course with high score, etc.)"*—Policy Maker 1; one instructor responded that: *"I am surprised by the students' creativity in finding new ways during COVID-19 in cheating within online examination activities"*—Instructor 4; *"Current online procedures do not monitor and control neither the physical nor the digital context in which the student participates during online examinations"*—Instructor 3; *"The success rate in subjects has risen in the last year, probably because the students adapted to the new systems and found loopholes. Oral examinations are harder to cheat"*—System Administrator 3.

We asked participants how much they trusted the process, in terms of whether the grades received by students actually reflected performances.

Representative responses from participants included the following: *"The grades don't reflect the performance of the students. Students are still adjusting to the remote environment, and we don't have the tools to guarantee the fairness of the process"*—Instructor 1; *"I trust the process for the students that are already high-performing students, but I don't trust the process for the students that were low-performing"*—Policy Maker 1; *"Mean value of grades hasn't changed much. However, there is no evidence that students didn't commit some fraud in the exams"*—Instructor 2; *"Some students got a better grade than what they deserved. Some students had lower marks in in-person evaluations than in remote evaluations"*— Instructor 5; *"I trust the grading process if the number of students is up to 15 students— bigger classes are harder to monitor"*—Instructor 6; *"The system is not completely trustworthy. The grade in general might not reflect the performance"*—Student 5; *"We have no control over what they are doing during examinations, and it is difficult to countermeasure this without protecting their data"*—System Administrator 3.

**Main Observation.** The analysis of answers to *RQ1* revealed that there was a consensus among all participants/stakeholder groups that the current workflows and deployed Information and Communication Technologies (ICT) tools entailed vulnerabilities and, therefore, threatened the credibility of critical online academic activities, such as online examinations. These vulnerabilities should be identified and properly addressed to sustain credibility in procedures.

In more detail, participants' elaborations behind their responses revealed the absence of validated procedures in COVID-19 realms, compared to pre-COVID-19 validated procedures in which critical academic activities were conducted within controlled physical realms. In addition, current discrepancies in deployed ICT tools provide a vivid ground for students to think about "creative" methods and approaches to violate the codes of conduct and university policies in online examinations.

From a decision maker's perspective, responses revealed that all were very well aware of the limitations of the current examination methods and were working towards improving LMS features to address malicious activities, like plagiarism. One policy maker stressed that

the current online examination procedures entail a high number of threat scenarios, which make it very difficult to reach the standards of physical examinations. From the perspective of instructors, responses revealed emotional and ethical effects on instructors as they were not assured of fairness in the online examination procedures in distinguishing between students who were well prepared for the examinations and students who misused the limitations of the online examination procedures currently applied. From the perspective of students, all students questioned the current procedures within critical online academic activities, but, nonetheless, some students mentioned that the online-based procedure was easier and more convenient than conventional physical examinations.

### 3.3.2. Identification of Impersonation Threat Scenarios (*RQ2*)

We asked participants to report on their experiences, which related to threats with regards to student identification and verification during critical academic activities. Accordingly, we identified several impersonation threat scenarios, which are scenarios wherein a person imitates or replicates the behavior or actions of another person [29].

Representative responses from stakeholders included the following: *"I know for certain that at least 10 of my co-students paid external graduate students to impersonate them during the online examination of COVID-19"*—Student 2; *"Based on my experience, I had a class of 40–50 students for which I had to remember their faces in order to assure that no impersonation case occurred during the examination. This is not human possible"*—Instructor 1; *"I have concerns about exam integrity. You can't verify the identity of the students"*—Instructor 4; *"I don't trust the process. I am convinced that students are interacting with each other most of the time"* —Instructor 5; *"Regarding remote examinations, I am sure that students cheated in exams, and the grades improved a lot in the remote exams"*—Instructor 7; *"Regarding online exams, I believe it was a total failure, especially regarding lack of trustworthiness. To cope with this aspect, many instructors have put limits in the time of the exam to make it difficult for students to cheat, since they wouldn't have enough time to ask for help. Nevertheless, there is no evidence that a person who correctly verified their identity did not allow another person sitting next to them to take over and answer the questions of the examination"*—Policy Maker 2; *"During the written exams students could have committed a fraud. The oral examinations are more fair"*—Student 6; *"You cannot copy 100% of the exam but you can do a big part of it"*—Student 7.

We asked participants to elaborate on specific use cases in which they experienced impersonation behavior by students. Representative responses from participants included the following: *"I had an incident where a student had a certain dialect, which was different from the actual student so he got caught"*—Instructor 2; *"We couldn't be sure who actually took the exam. For example, a student could identify correctly himself initially, but there is no evidence that he was the one who actually took the exam and not someone else"*—Instructor 3; *"After going through the authentication logs of the LMS platform, it is evident that there were some cases that users were logged in concurrently from different internet addresses"*—System Administrator 2.

**Main Observation.** The analysis of answers to *RQ2* revealed several impersonation threats, which refer to cases in which a person other than the legitimately enrolled student takes the examination, and, consequently, submits/provides answers for the examination. There was a consensus among stakeholders of all three participating universities that online examinations suffered from impersonation activities that were identified throughout all online examination types, i.e., oral, digital written and hand written. In addition, responses revealed that the participating universities did not have adequate infrastructure and technologies in place to assure the credibility of the examinations. Major issues related to continuous student identification and verification aspects during online examinations.

We classify, in Table 3, the stakeholder responses. We identified the following impersonation threat scenarios: *(i)* a scenario in which a person imitates the student identification proofs (e.g., by changing the photograph on the student's identity) during the student identity verification process; *(ii)* a scenario in which the student switches seats with another person after the student identity verification process has been completed; *(iii)* a scenario

that embraces non-legitimate access to the Learning Management System of the university by third persons through sharing of identity management credentials.

**Table 3.** Summary of impersonation threats and threat scenario descriptions.

| Impersonation Threats | Threat Scenario Descriptions |
| --- | --- |
| Student violating identification proofs | A student changes the photograph on the identity card with the imposter's photograph or changes the student details on the identity card |
| Student switching seats after identification | A student is correctly identified and verified, and then switches seats during the examination session with an imposter |
| Non-legitimate person provides answers either digitally or hand written | Another, non-legitimate, person is concurrently logged in the LMS and provides answers either digitally or hand written, or uploads examination material in general |

### 3.3.3. Identification of Communication, Collaboration and Resource Access Threat Scenarios (*RQ3*)

We asked participants to identify important threat scenarios, based on their previous experiences in online examinations. Accordingly, we identified threat scenarios relating to communication and collaboration activities among students and other persons, and forbidden resource access by students, during online examinations. *Communication* refers to an act in which a student exchanges information with another person or persons by speaking, writing or using another type of medium. Communication scenarios may involve *computer-mediated communication threat scenarios*, in which a student communicates with another person through technological means and mediums, i.e., through video/audio conferencing tools, or in-situ *communication threat scenarios*, which take place within the student's physical context, in which the student communicates with another person through speaking and/or writing within the same physical context.

Furthermore, *collaboration* refers to an act in which a student works with another person or persons to produce a given result and achieve a certain goal. Collaboration activities can be either conducted through technological means and methods, i.e., through remote access and remote control computer software, or directly within the student's physical context. We refer to the former as *computer-mediated collaboration threat scenarios*, and the latter as in-situ *collaboration threat scenarios*.

Finally, *resource access threat scenarios* involve suspicious activities in which a student attempts to gain access to either digital and/or hard copy resources and material (e.g., books, slides, search engines), which are not to be accessed during an online examination, based on the examination's policy.

With regards to communication activities during online examinations, students mostly used an instant messaging application through a second monitor or were in the same room with fellow students who were taking the same examination.

Representative quotes from stakeholders included the following: *"Students were communicating during the examination via instant messaging applications. Usually they had two monitors. The first monitor was the one used for taking the examination, while the second monitor was used for having the instant messaging application open"*—Student 2; *"Some students were using shared Google Docs to share answers, Skype for business was not a good experience for online education. In one class, a student was acting up and the video was uploaded to YouTube"*—System Administrator 1; *"Students were mostly using their smartphone device to communicate with fellow students during the examination"*—Student 4; *"We have heard from fellow students that they were in the same place during the examination, sitting opposite to each other, and communicating orally since the microphone was muted. They told us that they were hiding their facial expressions while talking in order to avoid raising suspicions"*—Student 5; *"Students were mostly communicating via instant messaging applications on a second monitor"*—Student 7; *"There was lack of control during the examination and I believe that students were communicating via instant messaging*

applications"—Instructor 1. *"I had a case during an oral examination in which each time I asked the question to a specific student, the student delayed the answers for about 15–20 s and then provided the correct answer. Each time the student provided the answers, it seemed that she was reading out the answer and was not really aware what was she was reading"* —Instructor 3; *"Students can think of weird ways to cheat, especially considering that there is lack of proper control during the online exam. For example, I could have easily created a pre-recorded video of myself and switch the laptop's camera to display the pre-recorded video while I take the exam"*—Student 2.

With regards to collaboration activities during online examinations, students mostly collaborated remotely because it was more convenient and less risky than being in the same place. Nevertheless, there were also cases of physical collaboration.

Representative quotes from stakeholders included the following: *"In some examinations, the time of submitting each answer was not very strict. Therefore, students had enough time to ask a fellow student to connect to their computer via remote connection software, such as, TeamViewer, Remote Desktop Connection, and carry on the examination"*—Student 3; *"In some examinations we had to scan the answers we wrote on the paper and send them as PDF. In these cases, students were sending each other the scanned PDF via instant messaging applications"*—Student 4; *"Although some sections of the examination, or some questions within the same section were provided in random order, we had enough time to take a screenshot of the question and the answer, and send them via instant messaging to a group of twenty fellow students. This way, despite the random order of the sections/questions, we increased the chance of finding another person in this group that was assigned the same set of questions, and in the end, we all had access to all answers"*—Student 5; *"Being in the same place is more complex and risky than collaborating remotely, because in some examinations instructors asked students to first show their surroundings before starting the examination. Some students were not willing to take such a risk. However, there were some students that were in different rooms initially, and after showing their surroundings they sat next to each other and were able to collaborate"*—Student 6; *"There was a case in which students gathered in a lab room and a student connected to the projector. Then, during the examination, the student was providing the answers through the projector and the fellow students were simply copying the answer"*—Student 7; *"There was a case in an online oral examination in which the student had the microphone muted in each question, followed by a delay of ten-twenty seconds during which another person was providing the answer to the student orally"*—Instructor 3.

With regards to access to material that was not allowed, based on the examination's policy, students again mostly used a second monitor in which they had previously opened their notes or an instant messaging application in which they kept some notes. In some other cases, due to the nature of the examination, some instructors used a more controlled environment that prevented access to other applications.

Representative quotes included the following: *"Students were using a second monitor in which they had their notes opened"*—Student 2; *"In some examinations the environment was locked so we couldn't have access to our material or search on the Internet. However, in many other examinations this wasn't the case"*—Student 4; *"Although some instructors were hiding the material in the LMS, students were still able to download all the relevant lecture notes in advance and have them opened during the examination"*—Student 6; *"Some instructors decided to use more controlled environments during the online examinations through additional tools, depending on the nature of the examination. However, there were many cases in which the installation and setup of the software was difficult and frustrating to students"*—Instructor 2.

**Main Observation.** The analysis of the answers to *RQ3* revealed several threats that relate to communication activities, collaboration activities, and gaining access to material that is not allowed during online examinations. We identified the following computer-mediated communication, collaboration and resource access threat scenarios (classified in Table 4): *(i)*: a scenario in which computer-mediated communication happens through voice or text-written chat, using either the same computing device as the one used for the examination or another computing device, or in cases where another person is co-listening and/or co-viewing the examination question, and, then, provides answers through text-written or voice communication, either using the same computing device as that used for

the examination, or with another computing device; *(ii)*: a scenario in which the computer-mediated collaboration happens through remote access, with control and screen sharing applications using either the same computing device as the one used for the examination, or another computing device; and *(iii)*: a scenario in which a student is finding help from online resources, search engines, hard copy material, not allowed in the examination policy, either using the same computing device as the one used for the examination, or another computing device.

**Table 4.** Summary of computer-mediated communication, collaboration and resource access threats and scenario descriptions.

| Computer-Mediated Communication, Collaboration and Resource Access Threats | Threat Scenario Descriptions |
|---|---|
| Computer-mediated communication through voice or text-written chat | A student remotely communicates with another person through voice or text-written chat, either using the same computing device as the one used for the examination, or with another computing device. Alternatively, another person co-listens to an examination question within an oral examination, and, then, provides answers through written text or voice communication, either using the same computing device as the one used for the examination, or another computing device |
| Computer-mediated collaboration through screen sharing and control applications | A student remotely communicates with another person through special applications (e.g., share screen, remote desktop connection), either using the same computing device as the one used for the examination, or another computing device |
| Student access to forbidden online resources | A student finds help from online resources, search engines, forbidden by the examination policy, either using the same computing device as the one used for the examination, or another computing device |

Furthermore, we identified the following in-situ communication, collaboration and resource access threat scenarios (classified in Table 5): *(i)*: a scenario in which another person provides answers through the student's main input device on the student's computer, or through a secondary input device; *(ii)*: a scenario in which a student communicates/collaborates with another person within the same physical context; and *(iii)* a scenario in which another person projects answers through a white board/computing device/hard copy messages.

**Table 5.** Summary of in-situ communication, collaboration and resource access threats and scenario descriptions.

| In-Situ Communication, Collaboration and Resource Access Threats | Threat Scenario Descriptions |
|---|---|
| Non-legitimate person provides answers on the student's computing device through the main, or a secondary, input device | A student sits in front of the camera, and a non-legitimate person sits next to the student, typing the answers through the student's main input device or through a secondary device (keyboard, computer mouse, etc.) displayed on the student's computer screen |
| Student communicating/collaborating with another person within the same physical context | A student communicates/collaborates (i.e., talks) with another person, that is not in the field of view of the camera, within the same physical context |
| Non-legitimate person provides answers through a white board/computing device/hard copy messages | A non-legitimate person provides answers through a computer and projects the answers using a white board/computing device/hard copy messages |

## 4. Rating of Threat Scenarios

In this section, we rate the identified threat scenarios in terms of likelihood/probability of occurrence and their levels of severity, keeping in mind the type of online examination (oral, digital written, hand written). As such, our aim was to highlight the most probable and severe threats for which countermeasures need to be identified and applied during the implementation of an online examination. To do so, we asked participants about the probability and severity of each threat scenario, contextualized within each online

examination type. With regards to probability, participants responded by indicating one option out of four probability levels (High/Medium/Low/Not Applicable), and, with regards to severity, participants responded by indicating one option out of four severity levels (Major/Medium/Minor/Not Applicable).

### 4.1. Sampling and Procedure

A total of seven (7) participants were recruited from three European universities. The sample included participants with a variety of roles, i.e., four (4) instructors, one (1) decision maker, 1 (one) system administrator and one (1) data protection expert. We conducted a series of semi-structured interviews with each participant that lasted approximately 60–90 min each, discussing topics related to the rating of the threat scenarios identified in terms of likelihood/probability of occurrence and level of severity.

### 4.2. Rating of Impersonation Threats

Table 6 summarizes the participants' ratings of impersonation threat scenarios per examination type (oral, digital written, hand written). Accordingly, in oral online examinations, the most probable impersonation threat scenario related to students imitating the identification proofs, which was also indicated as the most severe threat in cases of this scenario occurring. The scenario related to students switching seats after successful identification was less probable, given that students are closely monitored in real-time by the instructor, while the scenario in which a person submits answers through shared LMSs was, in most cases, not applicable. One participant stated that *"When you are doing the oral examination you are being watched in a close way, so it is not easy to switch seats after the student has been identified"*—Instructor 1.

**Table 6.** Impersonation threat ratings per examination type (responses from seven participants).

| Impersonation Threats | Oral | | Digital Written | | Hand Written | |
|---|---|---|---|---|---|---|
| | Likelihood | Severity | Likelihood | Severity | Likelihood | Severity |
| Student violating identification proofs | **High** **(7)**; Medium (0); Low (0) | **Major** **(7)**; Medium (0); Minor (0) | High (2); Medium (5); Low (0) | Major (6); Medium (1); Minor (0) | High (1); Medium (5); Low (1) | Major (6); Medium (1); Minor (0) |
| Student switching seats after identification | High (0); Medium (1); Low (6) | Major (6); Medium (1); Minor (0) | High (1); Medium (3); Low (3) | Major (6); Medium (1); Minor (0) | High (1); Medium (3); Low (3) | Major (4); Medium (2); Minor (1) |
| Non-legitimate person provides answers either digitally or hand written | N/A | N/A | **High** **(6)**; Medium (1); Low (0) | **Major** **(7)**; Medium (0); Minor (0) | **High** **(6)**; Medium (1); Low (0) | **Major** **(6)**; Medium (1); Minor (0) |

With regards to both digital written and hand written online examinations, the most probable impersonation threats with major severity related to a non-legitimate person providing answers through shared LMS credentials. As participants responded, the reasoning behind this rating related to the fact that it is much easier to accomplish this activity compared to other threat scenarios. Nonetheless, other possible impersonation threat scenarios, which were stated to have medium probability of occurrence with major severity related to a student imitating identification proofs, and a student switching seats after successful identification. One participant responded that *"It is much easier for a student to share credentials with another person who will respond the answers in the LMS"*—Instructor 3.

**Main Observation.** All participants agreed that all impersonation threats are severe. However, depending on the examination type, the participants rated the attack probability differently, based on the ease of conducting each corresponding impersonation attempt by students. In *oral online examinations*, participants indicated that the scenario in which a student violates identification proofs is the most probable scenario to occur, whereas in *digital written and hand written online examinations*, all threat scenarios are more likely

to happen, given that the student is not closely monitored, as in the case of the oral examinations. Nonetheless, participants indicated that the highest probable scenario to occur in *digital written and hand written online examinations* related to a non-legitimate person providing answers either digital, through shared LMS credentials, or hand written.

### 4.3. Rating of Communication, Collaboration and Resource Access Threats

**Computer-Mediated Communication, Collaboration and Access Threat Scenarios.** Table 7 summarizes the participants' ratings of computer-mediated communication, collaboration and resource access threat scenarios.

**Table 7.** Computer-mediated communication, collaboration and resource access threat ratings per examination type (responses from seven participants).

| Computer-Mediated Threats | Oral | | Digital Written | | Hand Written | |
|---|---|---|---|---|---|---|
| | Likelihood | Severity | Likelihood | Severity | Likelihood | Severity |
| Computer-mediated communication through voice or text-written chat | **High (7);** Medium (0); Low (0) | **Major (6);** Medium (1); Minor (0) | **High (6);** Medium (1); Low (0) | **Major (5);** Medium (2); Minor (0) | **High (6);** Medium (1); Low (0) | **Major (6);** Medium (1); Minor (0) |
| Computer-mediated collaboration through screen sharing and control applications | High (0); Medium (3); Low (4) | Major (3); Medium (2); Minor (2) | **High (5);** Medium (1); Low (1) | **Major (5);** Medium (1); Minor (1) | High (1); Medium (2); Low (4) | Major (3); Medium (2); Minor (2) |
| Student access to forbidden online resources | High (3); Medium (3); Low (1) | Major (7); Medium (0); Minor (0) | **High (5);** Medium (1); Low (1) | **Major (6);** Medium (1); Minor (0) | High (3); Medium (2); Low (2) | Major (5); Medium (0); Minor (2) |

**Main Observation.** In *oral online examinations*, the most probable and severe computer-mediated threat scenario related to students communicating either through voice or text-written chat with another person. With regards to *digital written online examinations*, participants rated all computer-mediated threat categories to be highly probable to occur, with most of the participants indicating major severity. The main reason relates to the fact that, during digital written examinations, students may concurrently use the same medium or different computing devices (e.g., a smartphone) to receive help during the online examination. With regards to *hand written online examinations*, participants rated computer-mediated communication through voice or text-written chat as the threat with the highest probability to occur, with major severity. Computer-mediated collaboration in hand written online examinations is more difficult to conduct, given that the student is not primarily using a computing device to write the answers, and, hence, such suspicious behavior might be easily identified by instructors/invigilators.

**In-situ Communication, Collaboration and Resource Access Threat Scenarios.** Table 8 summarizes the participants' ratings of in-situ communication, collaboration and resource access threat scenarios.

**Main Observation.** In *oral online examinations*, the most probable and severe in-situ threat scenario related to students receiving answers through projections on a white board, computing device and/or in hard copy messages from a person in the same physical context as the student. In addition, some participants rated a student communicating with another person within the same physical context as being more difficult, given the synchronous interaction and monitoring of the student during oral examinations. With regards to *digital written online examinations*, participants rated all in-situ threat categories (in-situ communication, collaboration, resource access) as highly probable to occur, with most of the participants indicating these with major severity. On the other hand, for *hand written online examinations*, the most probable threat was rated to be the scenario in which another person provides answers through a white board/computing device/hard copy messages.

**Table 8.** In-situ communication, collaboration and resource access threat ratings per examination type (responses from seven participants).

| In-Situ Threats | Oral | | Digital Written | | Hand Written | |
| --- | --- | --- | --- | --- | --- | --- |
| | Likelihood | Severity | Likelihood | Severity | Likelihood | Severity |
| Non-legitimate person providing answers on the student's computing device through the main or secondary input device | High (3); Medium (3); Low (1) | Major (3); Medium (2); Minor (2) | **High (6)**; Medium (1); Low (0) | **Major (6)**; Medium (1); Minor (0) | High (3); Medium (2); Low (2) | Major (3); Medium (3); Minor (1) |
| Student communicating/ collaborating with another person within the same physical context | High (1); Medium (1); Low (5) | Major (3); Medium (2); Minor (2) | **High (6)**; Medium (1); Low (0) | **Major (6)**; Medium (1); Minor (0) | High (3); Medium (2); Low (2) | Major (3); Medium (2); Minor (2) |
| Non-legitimate person providing answers on a white board/computing device/ hardcopy messages | **High (7)**; Medium (0); Low (0) | **Major (6)**; Medium (1); Minor (0) | **High (6)**; Medium (1); Low (0) | **Major (6)**; Medium (1); Minor (0) | **High (5)**; Medium (2); Low (0) | **Major (5)**; Medium (2); Minor (0) |

## 5. Threat Model, Data Metrics and Countermeasures

In this section, we present suggested countermeasures and required data metrics to address the identified impersonation threats, and communication, collaboration and resource access threats (Figure 1). Important data metrics relate to students' biometric data, i.e., physiological and behavioral, to primarily address impersonation threats, and to contextual data to address suspicious behavior with regard to communication, collaboration and resource access threats. We also discuss issues of privacy preservation in storing, retrieving and processing biometric data, and state-of-the-art approaches and technologies to overcome such issues. Appendix B summarizes the countermeasures and features for each threat.

### 5.1. Addressing Impersonation Threats through Physiological Data Analysis

Impersonation threats can be addressed by analyzing students' biometric data (physiological and behavioral), such as face and voice data. Specifically, impersonation threats during *student examination enrolment* can be addressed through automatic student verification, based on ground truth biometric data, while impersonation threats during an *examination session* can be addressed by means of continuous student identification, based on biometric data. In addition, impersonation threats can be identified, *after an online examination* has been completed, through intelligent data analytics, based on historical biometric data.

**Leveraging Ground Truth Physiological and Behavioral Data to Verify Students.** This countermeasure aims to address the impersonation threat scenario that relates to a *student violating identification proofs*, relevant to *all online examination types (oral, digital written, hand written)*. Specifically, it aims to perform the student identification process through three main automated functionalities: *(i) face-based identification:* comparison of the student's face characteristics and the identity card provided with ground truth data provided by the university; *(ii) voice-based identification:* comparison of the student's voice signals and ground truth voice data provided by the university; and *(iii) knowledge-based identification:* asking the students a series of secret questions (e.g., what is your grandmother's name) and comparing the answers with ground truth data. Furthermore, for student verification, i.e., verifying that a student is enrolled on the instructor's student examination list. This functionality includes automated comparison of the student's photograph and full name on the student's identity card with ground truth data (student's photograph and full name)

provided by the university and face characteristics retrieved and analyzed during the aforementioned face-based identification approach.

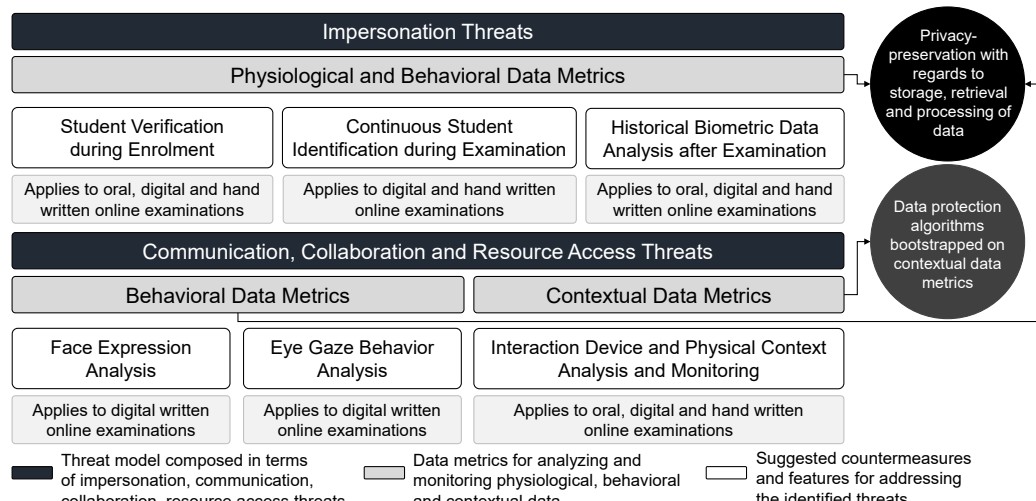

**Figure 1.** Threat model, data metrics and countermeasures for addressing impersonation threats, and communication, collaboration and resource access threats.

**Continuous Student Identification based on Physiological and Behavioral Data.** This countermeasure aims to address the impersonation threat scenario that relates to a *student switching seats after identification*, and a *non-legitimate person providing answers through shared LMS credentials*, which are primarily relevant in *digital written* and *hand written online examination* types. Specifically, it aims to continuously identify the student by comparing the student's face and/or voice data with ground truth (historical) data from the university's LMS. To do so, the first step involves continuously scanning the student's face with the web camera and/or recording the student's voice signals from the microphone every $x$ seconds, where $x$ is specified by the university and/or instructor. Consequently, the system then compares the face and/or voice data with ground truth (historical) data, retrieved from the university's LMS and/or the student's identification system. Historical data is stored per examination session for future analysis and comparison. Finally, in case of continuous student identification processes failing, as a fallback scenario the system alerts the instructor to either manually identify the student and/or proceed with different types of examinations (e.g., oral examination).

Other countermeasures for continuous student identification, that could be applied, aim to: *(i) detect authentic vs. pre-recorded input video streams* by applying specific methods to detect the authenticity of video streams; *(ii) monitor the student's login sessions* by checking whether concurrent login sessions exist from the same student; and *(iii) monitor the student's interaction device* by checking whether the characteristics of the student device have changed during the examination session.

**Intelligent Data Analytics to Identify Impersonation Cases, based on Historical Physiological and Behavioral Data.** This countermeasure aims to address *all impersonation threat scenarios* by analyzing historical data and identifying impersonation cases over time. This may be achieved by comparing facial embedding of students across multiple examination instances in the system aiming to detect whether a user with the same facial characteristics has been identified within multiple user accounts. Specifically, the system performs offline data analytics to detect historically based impersonation cases.

Furthermore, this countermeasure aims to properly utilize historical data to detect common handwriting styles between different student submissions, i.e., by comparing a student's submitted hand-written examination with previously submitted examinations from that student. Specifically, in order to recognize the student's handwriting style, the system implements methods based on Artificial Intelligence, Computer Vision, and Pattern

Recognition. In particular, the system first acquires and detects characters in the student's submitted documents. Next, it converts characters into the embedding, and, consequently, it compares the embedding (e.g., based on Euclidean distance) to decide if two characters have the same, or a different, author.

*5.2. Addressing Communication, Collaboration and Resource Access Threats through Behavioral and Contextual Data Analysis*

Communication, collaboration and resource access threats can be addressed by analyzing the students' behavioral data, such as face expressions and eye gaze behavior data, as well as monitoring the student's computing device and physical context. Such threats are primarily applied during an examination session to detect whether students are communicating and/or collaborating with another person, and whether they are attempting to access forbidden resources.

**Analysis of the Student's Behavioral Data and Patterns.** This countermeasure aims to primarily address in-situ communication, collaboration and resource access threat scenarios that relate to a *student communicating/collaborating with another person within the same physical context*, and a *non-legitimate person providing answers through a white board/computing device/hard copy messages*. Specifically, this is achieved by monitoring the student's behavioral data as follows: *(i) face behavior and expression tracking:* through state-of-the-art facial and expression recognition algorithms to identify any suspicious activity (e.g., model the facial expressions of students that provide answers to the examination system vs. students that are idle and only look at the screen); and *(ii) eye gaze behavior tracking:* through real-time eye gaze analysis (e.g., in the case where a student frequently looks beyond the monitor during the examination session).

With regards to face expression tracking, the system continuously tracks the face behavior and expressions of the student, through the student's web camera, to monitor and report on the frequency of face movements beyond the boundaries of the student's monitor, and face expressions during the examination session to identify certain patterns of behavior (e.g., whether the student is solving a problem vs. just looking at the screen). Furthermore, with regards to eye gaze behavior tracking, the system continuously tracks the eye gaze fixations and behavior of the student, through the student's web camera, to identify whether the student is constantly looking at another device (e.g., looking at another computer, smartphone for chat, etc.). Specifically, the system monitors and reports on the frequency of gaze fixations on certain positions on the screen, as well as identifies whether the student looks beyond the monitor.

**Monitoring the Student's Computing Device and Physical Context.** This countermeasure aims to address *all computer-mediated communication, collaboration and resource access threat scenarios*, and the *threat scenario in which a non-legitimate person provides answers on the student's computing device through the student's main input device or a secondary input device*. Specifically, this is achieved by monitoring the student's computing device and/or the physical context in which the online examination takes place as follows: *(i) voice signal processing*, through environmental audio signal processing algorithms to detect cases in which students might be talking to other persons. In this case, the system may further process the speech signals and convert them to text, that the instructor then uses to look into cases in which a student frequently talks during the examination session; *(ii) monitoring and controlling communication/collaboration applications*, to prevent students communicating and/or collaborating with other persons through certain applications; *(iii) monitoring and controlling access to websites* to allow or restrict access to specific websites depending on examination protocol and policy; *(iv) keyboard keystroke and computer mouse click analysis* to process and identify any keyboard-related, and/or computer mouse-related activity; and *(v) identifying secondary input/output devices*, which may be used by other persons to type in questions to the examination system.

## 6. Feasibility Study of an Intelligent and Continuous Online Student Identity Management System: Implementation and User Evaluation

According to the identified data metrics and countermeasures, we designed and implemented a proof-of-concept intelligent and continuous online student identity management system, namely TRUSTID. In this section we present the design and implementation details of the main components of the TRUSTID system, focusing on addressing impersonation attacks based on face-based and voice-based student identification mechanisms. The source-code of the TRUSTID system is available open-source for the public under the TRUSTID code repository (https://github.com/cognitiveux/trustid, accessed on 14 May 2023).

### 6.1. Conceptual and Architectural Design of the Trustid Framework

At a high-level, the TRUSTID framework consists of two main components: *(i)* the TRUSTID client applications; and *(ii)* the TRUSTID server applications. TRUSTID client applications are the main source of student interaction and are responsible for capturing student face and voice samples that are sent to the server for processing and storage. In addition, we implemented a smartphone application that serves as a privacy-preserving biometric wallet, enabling students to control and share their biometric data with their University. Furthermore, the TRUSTID server is a Web application that exposes endpoints whereby the TRUSTID client applications can interact with, and exchange, data. Furthermore, the TRUSTID server includes trained models for face (image) and voice analysis.

**TRUSTID Server and Client Applications.** TRUSTID follows a client–server architecture with an Application Programming Interface (API) at the server side, aiming to expose a list of end-points to the TRUSTID client applications. The server-side Web API is implemented as a Django application in Python 3.7.4, using the Django REST Framework. For the deployment of the server-side Web API, NGINX is used, which serves as a reverse proxy and load balancer. For certain heavy and time-consuming tasks, such as image and voice processing, we used Celery, which is an asynchronous task queue, based on distributed message passing, as well as a message broker, namely RabbitMQ. For storage of the data, PostgreSQL was used, which is an open-source relational database management system. Finally, we used Docker technology to easily pack, ship, and run our Web API as a lightweight, portable, and self-sufficient container. The *TRUSTID client applications* are implemented as two native desktop applications for Microsoft Windows and Apple MacOS systems. The proof-of-concept client prototypes can be deployed in Microsoft Windows (version 8 or later), and implemented using Windows Presentation Foundation and C Sharp programming language. Apple MacOS (version 11.6 or later) was implemented using SwiftUI 2.

**Student Identification Mechanisms, based on Face and Voice Analysis.** Two major components of the TRUSTID framework include *facial and voice identification*. *Facial identification* is the task of making a positive identification of a face in a photograph or video image against any pre-existing database of faces. Our current solution used a pre-trained network that extracts 128-D face embedding, provided by dlib (C++/Python high performance toolkit for Machine Learning) and pre-trained using the Visual Geometry Group (VGG) Face dataset, along with an outlier detection algorithm coupled with a voting classifier, which use the Euclidean distance to estimate the similarity of the images between a user's ground truth data and the user to verify. *Voice identification* is the task of identifying or verifying someone's identity based on their voice characteristics. An open-source conversational AI toolkit called SpeechBrain (https://speechbrain.github.io, accessed on 26 May 2023) provided the foundation for the voice-based voice identification mechanism implemented in TRUSTID. It is intended to be straightforward, adaptable, and well documented. It performs admirably in a variety of fields.

**Privacy preserving biometric wallet.** We implemented a smartphone application enabling the end-user student to store, control and share biometric models (face and voice) with his or her University. Figure 2 illustrates representative screens of the application, including an application dashboard presenting the biometric data of the student shared

with the University, and a push notification functionality in which students approve or deny sharing biometric data in cases in which the University requests access to student data. The application was implemented, based on React Native, so as to deploy it in both Apple iOS and Google Android operating systems.

### 6.2. User Evaluation of TRUSTID

We conducted a user study aiming to evaluate the following: *(i)* the resilience of TRUSTID to impersonation attacks during an online examination by evaluating the implemented face- and voice-based identification mechanism; *(ii)* the resilience of TRUSTID in forbidden communication/collaboration scenarios; *(iii)* usability and user experience of end users, based on their interactions with the TRUSTID system; and *(iv)* perceived security and privacy of users in regard to the TRUSTID system.

**Sampling and Procedure.** A total of 133 individuals (female 29.4%; male 70.6%) participated in the user study with age groups ranging between 18 to 50 years old, with the majority of participants (71.6%) falling in the 18–24 age group. All individuals participated voluntarily and could opt-out from the study at any time. We adopted the University's human research protocol in regard to considerations of user privacy, confidentiality and anonymity. Participants were invited to join an online meeting at a time convenient for them. The user study was conducted with groups of up to 15 participants at a time. To evaluate the system, participants joined a Zoom call, and were guided by a researcher who also acted as the instructor of the examination. Participants were asked to download the TRUSTID software using a link that we provided, and to log in using credentials they received via email.

Two mock examination scenarios were evaluated with student participants across three different European countries: *(i)* **online written examination:** during the online written examination, the students were requested to answer to both multiple-choice and open-ended questions of generic knowledge. Overall, 65 students participated in the online written examination assessment, and the average time spent was 9.25 ± 5.23 min; and *(ii)* **online oral examination:** during the online oral examination, the students were requested to answer open-ended questions of generic knowledge asked by a researcher who acted as the instructor. Overall, 68 students participated in the online oral examination assessment, and the average time spent was 7.43 ± 6.01 min.

We followed a three-phase method in conducting the study, as follows:

*Phase 1—Face and Voice Biometric Registration:* Participants were instructed to enroll in the TRUSTID system by registering their face and voice samples. During this phase, participants were guided by the system in which their face and voice samples were recorded using the computer's Web camera and microphone, respectively. These biometric samples were used as ground truth data and stored in the end-user client-side application, enabling sharing of, and control of, the biometric data with the University during an examination.

*Phase 2—User Identification and Computer Monitoring:* Participants were instructed to attend the examination, by first completing the face and voice identification process, during which the system recorded, modeled and, then, compared the model with the ground truth model obtained during Phase 1. Additionally, the monitoring component checked participant running processes and applications in order that participants were prohibited from proceeding further with examination attendance in cases where a student wa using any forbidden communication/collaboration tools.

*Phase 3—Continuous Identification and Computer Monitoring during Online Examination:* Upon successful identification, participants attended an online examination, including a series of examination questions (multiple-choice, open-ended, etc.). Participants were also instructed to conduct a threat scenario (i.e., "cheat" the TRUSTID system, open a forbidden communication tool). Possible threat scenarios included impersonation attacks, e.g., the student switching seats with another person, as well as monitoring of forbidden communication/collaboration tools. Aiming to control and evaluate the resilience of the system to threat scenarios, participants were instructed to inform the system of an intention

to conduct the threat scenario by submitting their intentions through a feedback mechanism. This allowed us to compare what the system captured with what the participants attempted to do, aiming to evaluate the effectiveness of the implemented identification mechanisms. At the end of the study, participants completed a post-study questionnaire in which they provided feedback about the system, i.e., usability based on the System Usability Scale (SUS), user experience, and feedback on perceived security and privacy.

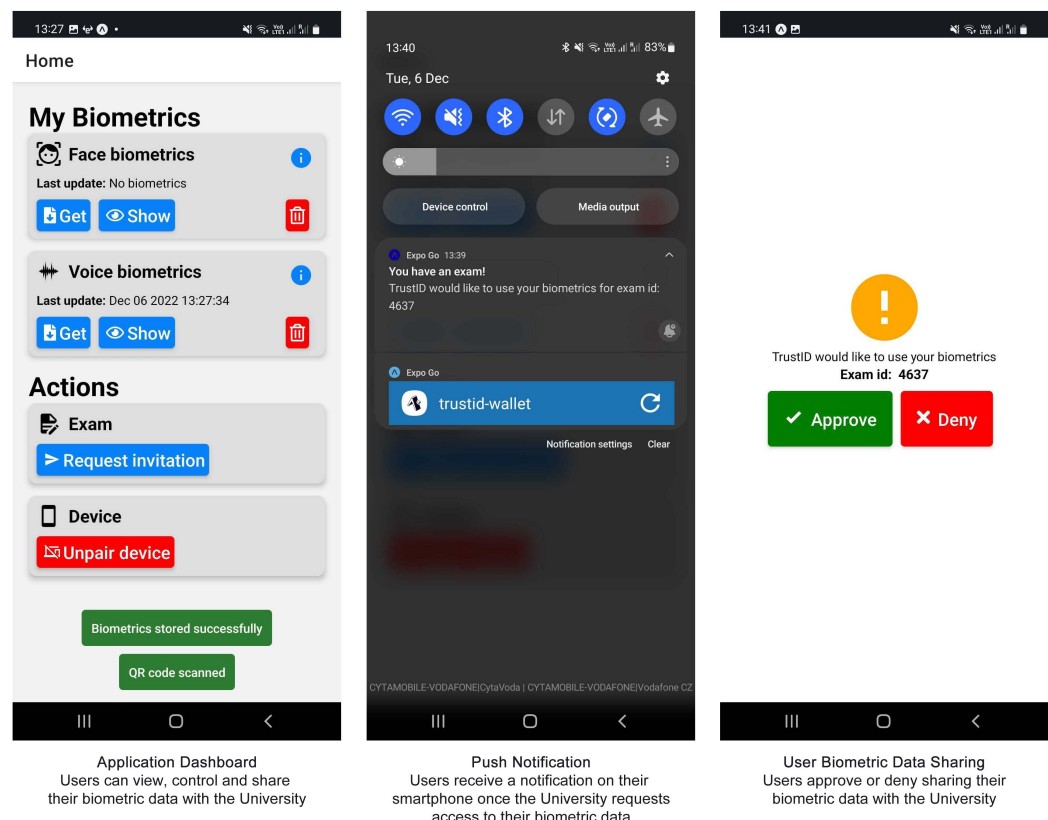

**Figure 2.** Snapshot of the privacy-preserving biometric wallet of TRUSTID.

### 6.3. Analysis of Results

**Resilience to Impersonation Attacks.** Once the student attended the mock online written examination, the monitoring system captured face images randomly every 5–8 s and audio samples every time the student spoke [30]. In the case of mock online oral examinations, the audio samples were captured upon the press-and-release of a button, in order to be able to separate the student's audio samples from the instructor's audio samples. Overall, from a sample of 133 participants, we collected 3334 face images and 119.15 min of audio samples, as shown in Table 9. From the sample of 133 students, we excluded 11 students from the voice recognition analyses only, since these students were not able to verify their voices and requested manual approval by the instructor to attend the examination.

**Table 9.** Summary of the sample and the collected data.

| Mock Examination Type | # of Participants | # of Face Images | Audio Samples Length (in minutes) |
|---|---|---|---|
| *Online Written* | 65 | 1804 | 75.68 |
| *Online Oral* | 68 | 1530 | 123.47 |
| **Totals** | **133** | **3334** | **199.15** |

Since we were interested in checking for similarities between the users' ground truth data and the data received in each identification attempt, we measured the accuracy of both face and voice recognition mechanisms by taking the number of correct identifications divided by the total number of identification attempts. Prior to attending a mock examination (either online written or online oral), students were also instructed to try to perform an impersonation attack if they wished (i.e., to try to cheat the face and voice recognition mechanisms). We intentionally did not tell students how to attempt to cheat but we encouraged them to use their creativity, technical knowledge, and imagination. Accordingly, we split the identification attempts into three cases, as follow:

*(i)* **Student identification in order to attend the examination.** In this case, the sample included the identification attempts of the legitimate students who were instructed to verify their faces and voices in order to proceed with attending the examination. The analysis of the results revealed that both the face recognition and the voice recognition mechanisms verified the students successfully 100% of the time.

*(ii)* **Continuous student identification prior to performing an impersonation attack.** In this case, the sample included the identification attempts of the legitimate students who continuously verified their faces and voices while participating in the examination, without attempting to launch any impersonation attack to cheat the system. The analysis of the results revealed that the face recognition mechanism verified the students successfully 94.80% of the time, while the voice recognition mechanism verified the students successfully 91.36% of the time.

To gain a better understanding of why the face and voice recognition mechanisms failed to verify the students successfully in some cases, we sporadically checked the raw data of some of the failed attempts. We observed that the face recognition failures mainly occurred due to face occlusion, inappropriate lighting conditions, and inappropriate posing of the head, while the voice recognition failures mainly occurred due to background noise and low amplitude of the signal.

*(iii)* **Continuous student identification while performing an impersonation attack.** In this case, the sample included the identification attempts of the students who attempted impersonation attacks to cheat the face and voice recognition mechanisms. Overall, from a sample of 133 participants, there were 56 participants who performed impersonation attacks, from which we collected 973 face images and 83.77 min of audio samples, as shown in Table 10. Each student was instructed to provide feedback prior to launching an impersonation attack. This was performed through the click of a button and a short description of the type of impersonation attack the student planned to launch. To isolate these cases, we used the timestamp of each impersonation attack attempt (which was provided by the student) and excluded all identification attempts of that individual prior to the impersonation attack timestamp. The analysis of the results revealed that the face recognition mechanism successfully detected that a student attempted an impersonation attack 76.57% of the time, while the voice recognition mechanism successfully detected that a student attempted an impersonation attack 78.53% of the time. Table 11 summarizes the results for each of the aforementioned identification cases.

**Table 10.** Summary of the sample and the collected data for impersonation attacks.

| Mock Examination Type | # of Participants | # of Face Images | Audio Samples Length (in minutes) |
|---|---|---|---|
| *Online Written* | 24 | 391 | 31.04 |
| *Online Oral* | 32 | 582 | 52.73 |
| **Totals** | **56** | **973** | **83.77** |

**Table 11.** Summary of the results for each identification case.

| Identification Case | Face Recognition (Success Rate) | Voice Recognition (Success Rate) |
| --- | --- | --- |
| *Student identification in order to attend the examination* | 100% | 100% |
| *Continuous student identification prior to performing an impersonation attack* | 94.80% | 91.36% |
| *Continuous student identification while performing an impersonation attack* | 76.57% | 78.53% |

The observed high percentages suggest the feasibility and the positives of the implemented face- and voice-based recognition mechanisms in automatically verifying students, based on face and voice characteristics, respectively. To gain a better understanding of why the face and voice recognition mechanisms failed to verify the students successfully in some cases, we sporadically checked the raw data of some of the failed attempts. We observed that the face recognition failures mainly occurred due to face occlusion, inappropriate lighting conditions, and inappropriate posing of the head, while the voice recognition failures mainly occurred due to background noise and low amplitude of the signal.

**Resilience to Forbidden Communication/Collaboration Scenarios.** The computer monitoring component periodically captured running processes and applications on the participants' interaction devices. The component correctly detected 100% of the cases in which participants indicated, through a feedback mechanism, an attempt to "cheat" on the examination using a forbidden communication/collaboration tool. Furthermore, as identified in the threat scenarios, prohibited communication and collaboration among students may entail more scenarios (e.g., student communicating/collaborating with another person within the same physical context). Such scenarios were out of the scope of the reported work and are to be the focus of future activities of this research endeavor.

Furthermore, Table 12 summarizes the user evaluation results with regards to user experience, perceived security and privacy. Out of 133 participants, 102 completely responded to the post-study questionnaire.

**Usability and User Experience Evaluation.** Aiming to evaluate the overall system usability of TRUSTID, we used the System Usability Scale (SUS), which is an accredited and widely applied instrument for usability evaluation. According to the participants' responses, the overall usability score of TRUSTID was 78.5%. Based on the literature, the average SUS score was 68% [31], with scores above that threshold indicating that a system entails good usability practices. Qualitative end-user feedback further supported the SUS score, e.g., when participants were asked *"Overall, how simple and clean is the TRUSTID software's user interface?"*, the majority (89 participants) indicated that they very much, or strongly, agreed, 10 moderately agreed, while 3 responded negatively. Participants also responded positively to the statement *"Overall, how intuitive to navigate is the TRUSTID software's user interface?"*, with 89 participants agreeing very much, or strongly, 11 moderately agreeing and 2 participants not agreeing. Furthermore, we asked participants *"Overall, what's your opinion on the way features and information in the TRUSTID software are laid out?"*, receiving positive responses as follows: 87 participants agreed very much, or strongly, 10 agreed moderately, while 5 participants did not agree with the statement. Indicative positive responses included the following: *"Overall it was very intuitive to use the software, I only had a few problems submitting feedback but nothing major, user identification was quick and overall it looks great!"*; *"The system is very easy to use, it can be really useful for professors that are not very comfortable with technology"*; *"Easy to use, fast to learn and useful for online exams"*.

**Table 12.** Summary of user evaluation results with regards to user experience, perceived security and privacy. The results indicate the total number of participants that disagreed/agreed with a specific statement.

| Question | Disagree | Moderate | Agree |
|---|---|---|---|
| Overall, how simple and clean is the TRUSTID software's user interface? | 3 | 10 | 89 |
| Overall, how intuitive to navigate is the TRUSTID software's user interface? | 2 | 11 | 89 |
| Overall, what's your opinion on the way features and information in the TRUSTID software are laid out? | 5 | 10 | 87 |
| Overall, how secure do you find the face identification process? | 9 | 22 | 71 |
| Overall, how secure do you find the voice identification process? | 12 | 23 | 67 |
| Overall, do you like the idea to be identified with face-based biometric identification during an online examination? | 21 | 20 | 61 |
| Overall, do you like the idea to be identified with voice-based biometric identification during an online examination? | 26 | 24 | 52 |

On the downside, while TRUSTID scored well in usability characteristics, there were some aspects requiring improvement. The qualitative user feedback indicated that the voice-based registration and identification process were, for some student participants, rather difficult and negatively affected usability and user experience. In addition, some participants indicated that they found the installation process relatively difficult, affecting overall user experience within the system. Indicative negative responses included the following: *"It is not available to linux users so I had to use another computer"*; "I had to disable Antivirus, because the program crashed during voice biometrics registration"; "Audio recognition default device was not my current mic input".

**Perceived Security and Privacy Evaluation.** During the post-study questionnaire, we further asked participants to rate their perceptions towards security and privacy. With regards to perceived security towards the face and voice identification process (*"Overall, how secure do you find the face identification process?"*; *"Overall, how secure do you find the voice identification process?"*, participants responded as follows: for the face identification mechanism, 71 participants found the process highly secure, 22 moderately secure, and 9 non-secure; for the voice identification mechanism, 67 participants found the process highly secure, 23 moderately secure, and 12 non-secure.

Furthermore, we asked participants their opinions on whether they liked the idea of using face and voice identification mechanisms during online examinations, receiving mixed responses from the participants. Specifically, 61 participants liked the idea of face-based biometric identification during an online examination, 20 moderately liked the idea, while 21 did not like the idea. Furthermore, 52 participants liked the idea of voice-based biometric identification during an online examination, 24 moderately liked the idea, while 26 did not like the idea.

## 7. Privacy Preservation Issues and Challenges in Storing, Retrieving and Processing Biometric Data of Students

The above-mentioned evaluation results are positive for further investigation and for improving the implemented TRUSTID framework, since the results showed that the implemented face and voice identification mechanisms were resilient to impersonation attacks, the TRUSTID system scored well in usability and user experience, and end users responded positively with regards to the perceived security and privacy of the implemented technology. Nonetheless, the suggested countermeasures primarily depend on physiological and behavioral biometric technologies, which currently face several issues related to the preservation of the privacy of sensitive personal biometric data [32,33]. To address aspects of the preservation of privacy of students' data, we envisioned a scenario in which students control access to their data by following self-sovereign data protection architectures [34–36], which allow end users to fully control who and what data are shared. In this context, it is an-

ticipated that universities act as trusted entities with certified procedures that keep ground truth biometric data from their students in order to assure effective and efficient student identification, verification and monitoring. Moreover, historical analysis over time requires the storage of large amounts of data about students. State-of-the-art approaches [37–41] include the following: biometric encryption techniques, such as homomorphic encryption [39,41]); protocol-based approaches, such as secure multiparty computation protocols, zero-knowledge proof protocols, etc. [41]; distributed ledger technologies, having certain features that can address several challenges associated with preservation of the privacy of biometric data. The single point of failure can be addressed by its distributed nature, elimination of third parties and potential privacy leakage, and monitoring and access to trusted and unalterable history logs [40,42–46].

In this context, we suggest the following state-of-the-art architectural solutions to address preservation of privacy issues of storing and processing sensitive biometric data of end users (Figure 3).

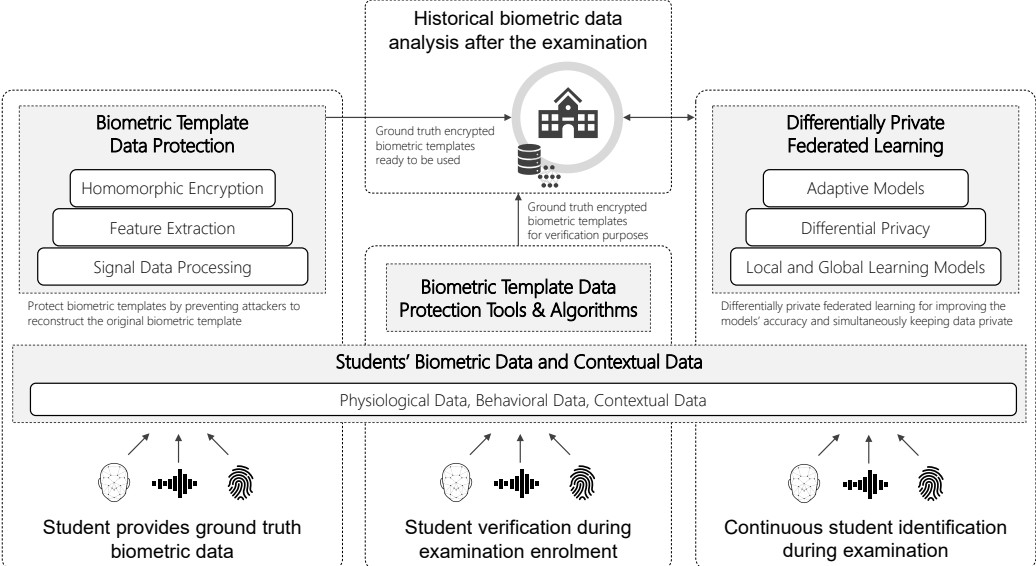

**Figure 3.** State-of-the-art architectural solutions for addressing preservation of data privacy.

**Privacy preserving Methods for Biometric Template Protection.** The primary step of biometric technology relates to student enrolment in the system, in which the system retrieves ground truth biometric samples (e.g., face samples, voice signals, fingerprint data, etc.). These raw samples are, then, processed, in order to construct a biometric template, a digital representation of unique features from the biometric sample [47,48], which is, then, communicated from the user's device and stored in a remote database. Unprotected biometric templates may lead to privacy breaches of biometric data, and compromised security of the authentication system, given that the raw biometric samples can be reconstructed by attackers based on the biometric templates. To address this issue, biometric technologies may apply biometric template protection methods based on neural networks-based transform approaches [49–52], and homomorphic encryption [53–55], etc. Several works focus on improving the performances of such methods (e.g., [54–56]), given that they are typically computationally expensive and practically infeasible, and require multiple user enrolments to improve the matching performance [54].

**Prevent Information Leakage from Trained Biometric Data Models, based on Federated Learning with Differential Privacy.** Another important step in biometric technology relates to training models based on users' biometric data, with large and diverse data for improved accuracy and performance. However, collecting, storing and processing such sensitive biometric data, and the respective trained data models, in a centralized manner increase privacy preservation issues. Hence, there is a need for methodologies to train biometric data without having full and direct access to the raw data and the trained

models. To prevent information leakage from trained biometric-based data models, several works have proposed approaches based on federated learning with differential privacy (e.g., [57–63]). Specifically, federated learning aims to aggregate biometric data from end users, to train the biometric-based data models across multiple decentralized computing machines (edge devices, servers, etc.), which hold data samples locally, and, then, to distribute a representative model to end-user devices. Through this process, the actual data models are not exchanged with the client devices. Further protective steps may be taken by including noise to the contributing data of each client during training, based on differential privacy methods.

**Scalable and Sustainable Biometric Data Storage Architecture and Data Sharing.** Biometric technologies entail interoperability issues for storing and sharing biometric data within and between different organizations and institutions given the heterogeneity of such biometric data and their respective data models. Hence, there is a need for sustainable and decentralized technologies for data storage and sharing, based on distributed ledger technologies, to increase scalability of biometric data [40,42]. In addition, there is a need for standardized data models that describe, in a holistic manner, static and dynamic contextual data and personal biometric-driven data that coherently reflects end users, based on semantic-based meta-data descriptions. Furthermore, there is a need for technologies that enable end users to control, manage and share their biometric data from their device, by applying a sustainable self-sovereign identity management approach [34,35]. To do so, a data privacy wallet smartphone application may be used by the end users to have full view and control over their shared data with their institutions. In addition, end users should be able to share their identity attributes (e.g., biometric data, sensitive personal data) securely over a decentralized system. Such an approach would allow organizations and institutions to request access to a particular user's data, shared only in cases in which the user wishes to share data without the need for a trusted third-party. For example, during an online examination, an instructor would request access to the student's biometric data, which would be shared only in cases in which the student wishes to share data.

## 8. Conclusions

This paper reports on the research results from a longitudinal study that spanned over eighteen months, and which aimed to identify, classify and rate threats and scenarios that depreciate the credibility of critical online academic activities. We also suggest countermeasures, features and relevant data metrics, which aim to address the threats. To the best of our knowledge, no other research work has identified and compiled a comprehensive list of threat scenarios and suggested countermeasures. These countermeasures were implemented under an intelligent and continuous student identity management system, which was evaluated with end users as to its resilience to impersonation attacks and its resilience to forbidden communication/collaboration scenarios. Usability aspects, user experience and perceived security and privacy of users towards the implemented technology were elicited. We increased the external validity of our study by recruiting numerous stakeholders with various backgrounds from three different European Higher Education Institutions (HEIs) (e.g., policy makers, instructors, data protection experts, administrators and students). We also increased the internal credibility and ecological validity of our study by conducting our research right after a three-semester period, during which the participating HEIs were obliged, due to the COVID-19 outbreak, to perform critical academic activities solely in online contexts. As such, we leveraged on timely real-life experiences from the participating stakeholders, which provided interesting and ecological valid insights in relation to our research questions. Evidence suggests that academic misconduct rose during the first year of the COVID-19 pandemic, and then fell during the second year [64], and, hence, it is probable that the prevalence of the studied threat scenarios might have changed over time.

The analysis of the results revealed that the participating HEIs followed a common pattern in conducting critical academic activities in online contexts (i.e., simultaneously utilizing Video Conferencing Tools and Learning Management Systems). However, this

strategy tolerated several malicious activities from students, which depreciated the credibility of student evaluation processes. Although the identified malicious activities were known from examinations in physical contexts (e.g., impersonation, forbidden communication/collaboration and access to forbidden resources), our research highlighted how these malicious activities evolved within online contexts. We also observed patterns of misconduct (threat scenarios) and rated these threat scenarios to each online examination type (e.g., oral, digital written, hand written).

From an academic perspective, being able to verify student identities and their continuous presence in critical online academic activities is of increased importance for the credible implementation of HEI curricula and for fair student evaluation processes. Our research revealed lessons learned from conducting critical academic activities in online contexts and envisions that the suggested countermeasures and features lead to tools that improve credibility of online critical academic activities.

From a technological perspective, the evaluation of the implemented TRUSTID framework indicated that the implemented face and voice identification mechanisms were resilient to impersonation attacks, since the majority of attacks were correctly identified by the system, and the system correctly identified legitimate and non-legitimate users. Furthermore, the post-study survey suggested that the TRUSTID system scored well in usability and user experience aspects, and end users responded positively with regards to perceived security and privacy of the implemented technology. Such results are encouraging for further study in, and improvement of, the implemented technology.

To this end, the identified threat scenarios, proposed countermeasures and implementation of open-source technologies could be a basis for broader contexts of online examination security, beyond the forced transition to online courses resulting from the COVID-19 pandemic. For example, the outcome of this work is relevant to HEIs offering distant learning courses and examinations, and online professional certification examinations, and enables traditional HEIs to offer a blended examination model (online and/or in-person examinations). From another perspective, the outcomes of this work may not only improve credibility and trustworthiness of online examinations, but also serve as a classroom attendance tool and student presence awareness tool. Being able to perceive, in a credible way, who is attending online synchronous and asynchronous scenarios, facilitates scaffolding and reproduces social situations that occur in the physical classroom, such as group attendance and awareness of classmates.

**Limitations and Challenges.** The limitations of this work relate to the rather small sample size in terms of participating universities and stakeholders. Nonetheless, the responses we received from the participants led to a consensus with regards to the investigated research questions. In addition, the implementation of the semi-structured interviews, especially in Phase B of the study (rating of threat scenarios), was rather cumbersome for participants in terms of required time, as we aimed to understand the reasoning behind responses. With regards to the implemented user identification mechanisms, in general, we observed, in our initial tests, that both face and voice identification mechanisms suffered from the generic issue of liveness [65,66] and non-authentic video streams [67]. As such, future work entails enhancing these mechanisms with liveness detectors that would be capable of detecting fake faces and voices, fake video streams and, thus, mitigate spoofing attacks.

Furthermore, the suggested countermeasures and features require the processing and/or storage of sensitive student data, such as facial embedding, voice signal streams, and interaction-related data. As such, a system implementing these features should be compliant with state-of-the-art privacy protection regulations (e.g., General Data Protection Regulation—GDPR) to preserve the privacy of student data [68–70]. In addition, the suggested countermeasures may raise privacy concerns, e.g., students may feel uncomfortable or stressed, due to run-time monitoring, and web cameras deployed in students' physical spaces may reveal personal/sensitive information. With regards to the former, future activities could investigate new designs and ways of informing and assuring students that

the monitoring is only to provide insights on academic misconduct, and that no sensitive/raw information is recorded. With regards to the latter, we would like to stress that the implemented technology crops the surroundings of the recorded image, and, instead, detects and further captures only the facial information of students.

Other future research directions of this work entail applying and investigating the implemented framework and technology in other domains, such as, the healthcare domain in combination with state-of-the-art user identification and knowledge-based user authentication mechanisms (e.g., [71–73]), and further enhancing the user identification mechanisms with complimentary modeling technologies and approaches, such as, eye tracking, cognitive modeling, etc. [74–76].

**Author Contributions:** Conceptualization, C.A.F., M.B., A.C. and D.P.; methodology, C.A.F., M.B., A.C. and D.P.; software, C.A.F., M.B., A.C., D.P. and P.M.; validation, C.A.F., M.B., A.C., D.P., P.M. and A.M.P.; formal analysis, C.A.F., M.B., A.C., D.P., P.M. and A.M.P.; investigation, C.A.F., M.B., A.C., D.P., P.M., A.M.P., A.P. and N.A.; resources, C.A.F., M.B., A.C., D.P., P.M., A.M.P., A.P. and N.A.; data curation, C.A.F., M.B., A.C., D.P., P.M. and A.M.P.; writing—original draft preparation, C.A.F., M.B., A.C., D.P., P.M. and A.M.P.; writing—review and editing, C.A.F., M.B., A.C., D.P., P.M. and A.M.P.; visualization, C.A.F., M.B., A.C., D.P., P.M. and A.M.P.; supervision, C.A.F., M.B., D.P., A.P. and N.A.; project administration, C.A.F., M.B., D.P., A.P. and N.A.; funding acquisition, C.A.F., M.B., D.P., A.P. and N.A. All authors have read and agreed to the published version of the manuscript.

**Funding:** The work has been partially supported by the European project TRUSTID—Intelligent and Continuous Online Student Identity Management for Improving Security and Trust in European Higher Education Institutions (Grant Agreement No: 2020-1-EL01-KA226-HE-094869), which is funded by the European Commission within the Erasmus+ 2020 Programme and the Greek State Scholarships Foundation I.K.Y. D. Portugal acknowledges the Portuguese Science Agency "Fundação para a Ciência e a Tecnologia" (FCT) for support under the Scientific Employment Stimulus 5th Edition, contract reference 2022.05726.CEECIND.

**Institutional Review Board Statement:** The security of Personal Data and the protection of all data subjects' rights and freedoms in relation to the processing of Personal Data under all the activities of the user studies conducted within the TRUSTID Project is fully compliant with the General Data Protection Regulation EU 2016/679 (GDPR) and the National Law 125(I)/2018.

**Informed Consent Statement:** Informed consent was obtained from all subjects involved in the study.

**Data Availability Statement:** Data is not available in a publicly accessible repository. The source-code of the implemented software technology (TRUSTID system) is available open-source for the public under the TRUSTID code repository (https://github.com/cognitiveux/trustid, accessed on 14 May 2023).

**Conflicts of Interest:** The authors declare no conflict of interest.

## Appendix A. Stakeholder Semi-Structure Interview Discussion Themes and Questions

*Appendix A.1. Prerequisites—Guidelines to the Interviewer*

The Interviewer should provide a brief description of the project and clearly state to the Interviewee the purpose of the interview, which is to gather information related to the validation goals. It is assumed that the Interviewee has already read a brief description about the project, otherwise the Interviewer cpnducts a brief introduction.

The Interviewer should mention that the data is handled anonymously. The recordings and responses are not shared with anyone beyond the partners of the project. Read the consent script to the Interviewee to inform her/him about the interview and to get her/his consent for recording of the interview. Start the recording.

*Appendix A.2. Discussion Themes and Questions*

Appendix A.2.1. Initial Profiling and Acquaintance (Approximately 5–10 min)

*Note: the purpose is to understand the Interviewee's background. This helps us to understand the context of her/his answers. The Interviewer should contextualize the word "trustworthy" in the context of the project: i.e., fairness of the academic process; reliability*

- Could you please tell us about your background and position in your organization? *[open-ended]*
- Which Learning Management System (LMS) does your University currently use? *[open-ended]*
- Which authentication types does your University currently deploy? *[open-ended]*
- What is the current authentication policy? *[open-ended]*

Appendix A.2.2. Experience and Trust with Regards to Critical Online Academic Activities (Approximately 30–40 min)

- Please inform us about the best and worst experiences you had with regards to critical online academic activities (examinations, laboratory work) during the COVID-19 period. *[open-ended]*
- Do you believe that the current procedure at your organization was trustworthy with regards to critical online academic activities (examinations, laboratory work) during the COVID-19 period? Justify your answer. *[Likert Scale (1–5): Not Trustworthy at All—Extremely Trustworthy; open-ended]*
- How much do you trust the process in terms of whether the grade a student receives is actually the grade reflecting performance? Justify your answer. *[Likert Scale (1–5): Not Trustworthy at All—Extremely Trustworthy; open-ended]*
- Share your experiences, relevant to the COVID-19 semesters, related to threats with regards to student identification and verification during critical academic activities. *[open-ended]*
- Identify, based on your previous experience in online examinations during the COVID-19 period, important threat scenarios. *[open-ended]*
- Elaborate on specific use cases in which you experienced impersonation behavior by students. *[open-ended]*

**Appendix B. Summary of Threats, Countermeasures and Features**

**Table A1.** Summary of identified threats, threat scenario descriptions, countermeasures and features.

| Identified Threats | Threat Scenario Descriptions, Relevant Countermeasures and Features |
| --- | --- |
| Student violating identification proofs | A student changes the photograph on the identity card with an imposter's photograph or the student changes details on the identity card.<br>*Countermeasure #1*: Student identification and verification; *Feature #1:* Face- or voice-based identification, and comparison of student's face characteristics with the picture on the student's identity card |
| Student switching seats after identification | A student is correctly identified and verified, and, then, switches seats during the examination session with an imposter.<br>*Countermeasure #2:* Continuous student identification; *Countermeasure #3:* Data analytics for historically based impersonation; *Feature #2:* Continuous scanning of the student's face characteristics, using the web camera, and/or recording the student's voice signals with the microphone; *Feature #3:* Detect authentic *vs.* pre-recorded input video streams; *Feature #6:* Perform offline data analytics to detect historically based impersonation cases; *Feature #7:* Comparison of student handwriting style with existing submitted handwriting style |
| Non-legitimate person provides answers either digitally or hand written | Another non-legitimate person is concurrently logged in the LMS and provides answers either digitally or hand written, or uploads general examination material.<br>*Countermeasure #2:* Continuous student identification; *Feature #4:* Monitoring the student's login sessions; *Feature #5:* Monitoring the student's interaction device |

**Table A1.** *Cont.*

| Identified Threats | Threat Scenario Descriptions, Relevant Countermeasures and Features |
|---|---|
| Computer-mediated communication through voice or text-written chat | A student is remotely communicating with another person through voice or text-written chat, either using the same computing device as the one used for the examination, or another computing device. Another person co-listens to the examination question, within an oral examination, and, then, provides answers through text-written or voice communication either using the same computing device as the one used for the examination, or another computing device. *Countermeasure #4:* Monitor student's digital context; *Countermeasure #5:* Monitor the student's behavior within the physical context; *Feature #8:* Monitoring and blocking communication and/or collaboration applications; *Feature #9:* Monitoring and blocking access to specific websites; *Feature #10:* Keyboard keystroke and computer mouse click analysis; *Feature #12:* Monitor voice signals, contextual sound and ambient sound; *Feature #13:* Face behavior and expressions analysis of the student; *Feature #14:* Eye gaze fixations and behavior analysis of the student |
| Computer-mediated collaboration through screen sharing and control applications | A student remotely communicates with another person through special applications (e.g., share screen, remote desktop connection), either using the same computing device as the one used for the examination, or another computing device. *Countermeasure #4:* Monitor student's digital context; *Countermeasure #5:* Monitor the student's behavior within the physical context; *Feature #8:* Monitoring and blocking communication and/or collaboration applications; *Feature #13:* Face behavior and expressions analysis of the student; *Feature #14:* Eye gaze fixations and behavioral analysis of the student |
| Student access to forbidden online resources | A student finds help from online resources and search engines, not allowed in the examination policy, either using the same computing device as the one used for the examination, or another computing device, . *Countermeasure #4:* Monitor student's digital context; *Feature #9:* Monitor and block specific websites |
| Non-legitimate person providing answers on the student's computing device through the student's main input device or a secondary input device | A student sits in front of the camera, and a non-legitimate person sits next to the student and types the answers through the student's main input device or a secondary device (keyboard, computer mouse, etc.), displayed on the student's computer screen. *Countermeasure #4:* Monitor student's digital context; *Countermeasure #5:* Monitor the student's behavior within the physical context; *Feature #10:* Keyboard keystroke and computer mouse click analysis; *Feature #11:* Check the drivers at the operating system level of the student's computing device; *Feature #13:* Face behavior and expressions analysis of the student; *Feature #14:* Eye gaze fixations and behavior analysis of the student |
| Student communicating/collaborating with another person within the same physical context | Happens when a student communicates/collaborates (i.e., talks) with another person that is not in the view field of the camera within the same physical context. *Countermeasure #5:* Monitor the student's behavior within the physical context; *Feature #12:* Monitor voice signals, contextual sound and ambient sound |
| Non-legitimate person providing answers on a white board/computing device/hard copy messages | A non-legitimate person provides answers through a computing device and projects the answers through a white board/computing device/hard copy messages. *Countermeasure #5:* Monitor the student's behavior within the physical context; *Feature #13:* Face behavior and expression analysis of the student; *Feature #14:* Eye gaze fixations and behavioral analysis of the student |

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
