# Peer review of "Ensuring Academic Integrity and Trust in Online Learning Environments: A Longitudinal Study of an AI-Centered Proctoring System in Tertiary Educational Institutions"

_education, doi:10.3390/educsci13060566_

Round 1
Reviewer 1 Report
Review Notes
Strengths:
· Important issue for online examination security in general, not just during a pandemic shift to online exams
· Good summary of challenges to online exam integrity.
· Descriptions of observations are clear.
· Data analyses are appropriate.
· Overall good organization.
· Design of the feasibility check was helpful. Challenges were interesting and plausible.
Recommendations:
· Clarify how this work applies more broadly to online exam security regardless of the forced transition to online courses resulting from the pandemic.
· Does this work address “prohibited communication and collaboration among students but also impersonation cases,” or just impersonation? It seems to focus on the threat of impersonation.
· Note that a limitation to Phases A and B is that the perceptions of stakeholders may not accurately reflect the level of threat. For example, much concern has focused on contract cheating in recent years, but the prevalence rates for other kinds of cheating and plagiarism are still much higher than for contract cheating. Does the research say anything about the prevalence of impersonation, and whether it has changed since the beginning of the pandemic? In fact, there is some evidence that concerns about Academic Misconduct rose during the first year of the pandemic, and then fell during the second year. See for example:
Wiley. (2022). New insights into academic integrity: 2022 update. 1–10. https://secure.wiley.com/academic-integrity-ebook
· Regarding Phase C, recognize that many recommendations have been made to reduce
Academic Misconduct. However, the research on the effectiveness of these approaches tends to be sparse and of poor quality. See for example:
Ives, B. & Nehrkorn, A. (2019). A Research Review: Post-Secondary Interventions to Improve Academic Integrity. In D. Velliaris (Ed.), Prevention and Detection of Academic Misconduct in Higher Education, Hershey, PA: IGI Global.
· Please provide more information about the “series of questions to elicit their perceived credibility of online academic activities.” How were the questions developed? Was an interview guide used?
· Perhaps avoid using the same labels (Phase A, Phase B, Phase C)for the three phases of the Methods, and also the three phases of the feasibility study.
Problems with English were minor and did not interfere with understanding.
Author Response
Dear Reviewer 1,
Thank you for your review and valuable feedback.
Please see the attachment, which includes our responses to your comments.
Kind regards,
The Authors

Reviewer 2 Report
I was very impressed with the background study, and the comprehensiveness of the threat analysis. This will be of great use to instructors in assessing the security of their own exams.
I do think, however, that the paper gives short shrift to privacy concerns, such as that the close monitoring suggested for student behaviors may cause students to become uncomfortable and may disadvantage students w/certain disabilities, and also that cameras deployed perhaps in a student's bedroom may pick up items that are embarrassing or compromising.
The threat ratings could be improved. Would it be possible to draw probabilities from hard data rather than a survey, especially considering the biases that may be involved when stakeholders themselves are asked to rate scenarios. Tables 6 to 8 were hard to read because of line wrap. It would be a lot clearer to list only the numbers and not "Major", "Medium", etc. And since all scores are out of 7, it's not necessary to repeat "/7" in every cell.
Having said this, the paper has convinced me sufficiently that their identified threats warrant attention; however, as a feasibility study (their main topic), I found their analytical methods to only be somewhat sufficient for now. The codebase that they cite (citation 7, used in page 16) actually leads to another paper, not to the open-source repository claimed in the paper. I do not know if this paper is a continuation of previous work, but I did find it to be lacking in the statistical rigor that their cited works display, especially when discussing examination scores and cheating probabilities.
Having said this, the paper has convinced me sufficiently that their identified threats warrant attention; however, as a feasibility study (their main topic), I found their analytical methods to only be somewhat sufficient for now. The codebase that they cite (citation 7, used in page 16) actually leads to another paper, not to the open-source repository claimed in the paper. I do not know if this paper is a continuation of previous work, but I did find it to be lacking in the statistical rigor that their cited works display, especially when discussing examination scores and cheating probabilities.
I am not an expert on systems like TRUSTID, and I hope that another reviewer familiar with those systems can vouch for the soundness of the work. I would guess that there must be many similar products, and don't know how thoroughly TRUSTID has been compared with the state of the art.
If I had more time for the review, I could have checked more thoroughly for background work. I do believe the authors have done the community a service by compiling a comprehensive list of threats, though I am not so sure how effectively TRUSTID addresses them.
Section 1 sounds stilted and contains errors such as use of infinitives where participles are required by idiomatic English, as well as errors in phrasal verbs (e.g., leaving out the preposition). After that, the English is much better, although in standard English, a student "begins an exam" or "takes an exam", rather than "joins an exam".
Author Response
Dear Reviewer 2,
Thank you for your review and valuable feedback.
Please see the attachment, which includes our responses to your comments.
Kind regards,
The Authors
